# SARS-CoV-2 ORF3a Protein as a Therapeutic Target against COVID-19 and Long-Term Post-Infection Effects

**DOI:** 10.3390/pathogens13010075

**Published:** 2024-01-14

**Authors:** Jiantao Zhang, Kellie Hom, Chenyu Zhang, Mohamed Nasr, Volodymyr Gerzanich, Yanjin Zhang, Qiyi Tang, Fengtian Xue, J. Marc Simard, Richard Y. Zhao

**Affiliations:** 1Department of Pathology, University of Maryland School of Medicine, Baltimore, MD 21201, USA; jiantao.zhang@som.umaryland.edu (J.Z.); chenyu.zhang@som.umaryland.edu (C.Z.); 2Department of Pharmaceutical Sciences, University of Maryland School of Pharmacy, Baltimore, MD 21201, USA; hom@rx.umaryland.edu (K.H.); fxue@rx.umaryland.edu (F.X.); 3Drug Development and Clinical Sciences Branch, Division of AIDS, NIAID, National Institutes of Health, Bethesda, MD 20892, USA; mnasr@niaid.nih.gov; 4Department of Neurosurgery, University of Maryland School of Medicine, Baltimore, MD 21201, USA; vgerzanich@som.umaryland.edu (V.G.); msimard@som.umaryland.edu (J.M.S.); 5Department of Veterinary Medicine, University of Maryland, College Park, MD 20742, USA; zhangyj@umd.edu; 6Department of Microbiology, Howard University College of Medicine, Washington, DC 20059, USA; qiyi.tang@howard.edu; 7Research & Development Service, VA Maryland Health Care System, Baltimore, MD 21201, USA; 8Department of Microbiology-Immunology, University of Maryland School of Medicine, Baltimore, MD 21201, USA; 9Institute of Human Virology, University of Maryland School of Medicine, Baltimore, MD 21201, USA; 10Institute of Global Health, University of Maryland School of Medicine, Baltimore, MD 21201, USA

**Keywords:** SARS-CoV-2, COVID-19, ORF3a, viral pathogenesis, cytokine storm, kidney injury, neuroinflammation, antiviral target, high-throughput screening

## Abstract

The COVID-19 pandemic caused by SARS-CoV-2 has posed unparalleled challenges due to its rapid transmission, ability to mutate, high mortality and morbidity, and enduring health complications. Vaccines have exhibited effectiveness, but their efficacy diminishes over time while new variants continue to emerge. Antiviral medications offer a viable alternative, but their success has been inconsistent. Therefore, there remains an ongoing need to identify innovative antiviral drugs for treating COVID-19 and its post-infection complications. The ORF3a (open reading frame 3a) protein found in SARS-CoV-2, represents a promising target for antiviral treatment due to its multifaceted role in viral pathogenesis, cytokine storms, disease severity, and mortality. ORF3a contributes significantly to viral pathogenesis by facilitating viral assembly and release, essential processes in the viral life cycle, while also suppressing the body’s antiviral responses, thus aiding viral replication. ORF3a also has been implicated in triggering excessive inflammation, characterized by NF-κB-mediated cytokine production, ultimately leading to apoptotic cell death and tissue damage in the lungs, kidneys, and the central nervous system. Additionally, ORF3a triggers the activation of the NLRP3 inflammasome, inciting a cytokine storm, which is a major contributor to the severity of the disease and subsequent mortality. As with the spike protein, ORF3a also undergoes mutations, and certain mutant variants correlate with heightened disease severity in COVID-19. These mutations may influence viral replication and host cellular inflammatory responses. While establishing a direct link between ORF3a and mortality is difficult, its involvement in promoting inflammation and exacerbating disease severity likely contributes to higher mortality rates in severe COVID-19 cases. This review offers a comprehensive and detailed exploration of ORF3a’s potential as an innovative antiviral drug target. Additionally, we outline potential strategies for discovering and developing ORF3a inhibitor drugs to counteract its harmful effects, alleviate tissue damage, and reduce the severity of COVID-19 and its lingering complications.

## 1. Introduction

The coronavirus disease 2019 (COVID-19) pandemic caused by SARS-CoV-2 infection was unprecedented in its rapid rate of spread, continuous emergence of new viral variants, and the high number of mortality cases and post-COVID complications. While vaccines have generally demonstrated efficacy in preventing SARS-CoV-2 infection or mitigating the severity of COVID-19, their effectiveness has often been compromised by the emergence of new viral variants and a decline in efficacy over time [1]. Promising antiviral drugs such as Paxlovid and molnupiravir have shown potential in treating COVID-19 and alleviating long COVID symptoms. However, challenges such as viral rebound and drug resistance have been observed [2,3]. Moreover, despite the pandemic showing signs of waning, a substantial number of infections persist worldwide, with the continued emergence of new viral variants. Given the relatively high mutable nature of coronaviruses (CoVs), there is an ongoing need to develop new antiviral drugs.

The development of antiviral drugs holds pivotal importance in the battle against COVID-19. Antiviral drugs play a critical role in various facets of disease management. For instance, remdesivir, an antiviral drug designed to impede SARS-CoV-2 replication, has demonstrated effectiveness in reducing the severity and duration of illness in COVID-19 patients [4]. Additionally, such medications diminish the viral load in infected individuals, thereby reducing the risk of transmitting the virus to others and aiding in community control. Antiviral drug development is essential in tackling emerging viral variants with diverse resistance profiles against existing treatments, necessitating ongoing research and development. Antiviral drugs can complement vaccination efforts, providing an added layer of protection in scenarios where vaccines are not universally available or effective due to the emergence of new viral variants. Finally, antiviral drugs constitute a crucial component of pandemic preparedness, aiding in readiness for potential future threats from emerging variants.

Numerous potential antiviral drug targets against SARS-CoV-2 and therapeutics against COVID-19 have been proposed and investigated. These encompass various approaches such as viral entry inhibitors targeting the spike protein or ACE2 receptor [5], protease inhibitors targeting the main protease (Mpro) like Paxlovid and papain-like protease (PLpro) [6] to impede viral protein processing, RNA-dependent RNA polymerase (RdRp) inhibitors such as remdesivir [7], ribonucleoside analogues such as molnupiravir, immunomodulators such as dexamethasone (a corticosteroid) [8] to manage excessive inflammation, monoclonal antibodies such as casirivimab and imdevimab (REGN-COV2) against specific viral components [9], and repurposed FDA-approved drugs such as hydroxychloroquine and ivermectin [8]. Clinical trials assessing their safety and efficacy have been pivotal, driven by the imperative to address the diverse symptoms and severity of COVID-19 cases, while also considering the potential impact of viral mutations on drug efficacy. Several of these drugs have obtained FDA authorization or approval for treating COVID-19. For instance, both Paxlovid and molnupiravir are FDA-approved. Remdesivir, an RdRp inhibitor, received emergency use authorization (EUA) and full approval for certain patient groups [4]. Casirivimab and imdevimab (REGN-COV2), monoclonal antibodies, obtained EUA for mild-to-moderate COVID-19 in high-risk individuals [9]. However, drugs like hydroxychloroquine and ivermectin, initially considered for emergency use, faced controversies due to a lack of sufficient evidence of efficacy, resulting in limited authorizations or revoked status by the FDA for COVID-19 treatment.

## 2. SARS-CoV-2 ORF3a Protein as a Promising Antiviral Target

To address the need for discovering novel antiviral targets against COVID-19, a comprehensive genome-wide functional analysis of SARS-CoV-2 proteins was conducted, utilizing a fission yeast cell-based system [10,11]. The selection criteria focused on identifying a viral therapeutic target crucial for viral pathogenesis associated with COVID-19 and exhibiting a quantifiable endpoint suitable for high-throughput screening (HTS), such as cell proliferation or cell death. Utilizing a fission yeast gene expression vector, a total of 29 SARS-CoV-2 viral proteins [12], covering the entire genome of the SARS-CoV-2 reference strain USA-WA1/2020 (GenBank#: MN985325) [13], were cloned [11,14,15]. Each viral gene was placed under an inducible *nmt*1 promoter [10,15,16], ensuring a specific measurement of the effects caused by the viral protein. Of the 29 SARS-CoV-2 viral proteins studied, the ORF3a (open reading frame 3a) protein uniquely met all the predefined selection criteria, emerging as a potential therapeutic target [17,18].

Increasing evidence underlines the clinical relevance of the ORF3a protein, as both SARS-CoV-1 and SARS-CoV-2-infected patients exhibited high levels of anti-ORF3a antibodies, and sera from recovering COVID-19 patients displayed considerable IgG and IgA reactivity specifically against ORF3a [19,20,21,22]. Notably, the IgG response to ORF3a was linked to severe COVID-19 across different age groups [23]. Robust SARS-CoV-2-specific T-cell responses targeting ORF3a were also observed in asymptomatic individuals or COVID-19 convalescent patients [24,25,26]. Furthermore, as described in detail in this review, ORF3a plays an important role in viral pathogenesis and contributes to disease severity of COVID-19. It is worth noting that numerous CoVs, including the seven known human CoVs (hCoVs), contain at least one accessory gene positioned after the S gene and before the E gene in their respective viral genomes. Some of these accessory proteins share similar functions, such as interference with ion channels and virion morphogenesis [27]. However, ORF3a, based on its overall protein structure, appears to be predominantly unique to SARS-CoV-2 [28], with its closest counterpart being SARS-CoV-1 ORF3a. This indicates a plausible recent adaptation of this protein both structurally and functionally during viral evolution [28]. The fact that both SARS viruses lead to severe human diseases, unlike other hCoVs, highlights the clinical significance of ORF3a in causing SARS or COVID-19.

## 3. Structure and Unique Features of ORF3a Protein

ORF3a is the largest accessory protein of SARS-CoV-2, displaying numerous functions that are not yet fully characterized (Figure 1A). Comparison of ORF3a protein sequences between SARS-CoV-1 and SARS-CoV-2 suggests that these two proteins share 73% sequence homology, with SARS-CoV-1 ORF3a also lacking one amino acid at position 241 (E241) [18,28]. Consequently, the SARS-CoV-2 ORF3a protein sequence is unique, and its three-dimensional (3D) tertiary structure is also novel, as it bears no resemblance to other known proteins [28,29].

Typically found as a homodimer or tetramer on the cell membrane [28], each monomer of ORF3a comprises 275 amino acids (aa), with a calculated molecular mass of 31 kD (Figure 1A). It harbors three hydrophobic transmembrane domains (TM1–TM3; aa40–128), anchoring it within the membrane. Both the N- and C-termini face the cytoplasm, housing distinct functional domains responsible for varied functionalities, including intracellular transport, ion channel formation and/or modulation, cytopathic effects, virus release, and virus production [18,29,30,31].

**Figure 1 pathogens-13-00075-f001:**
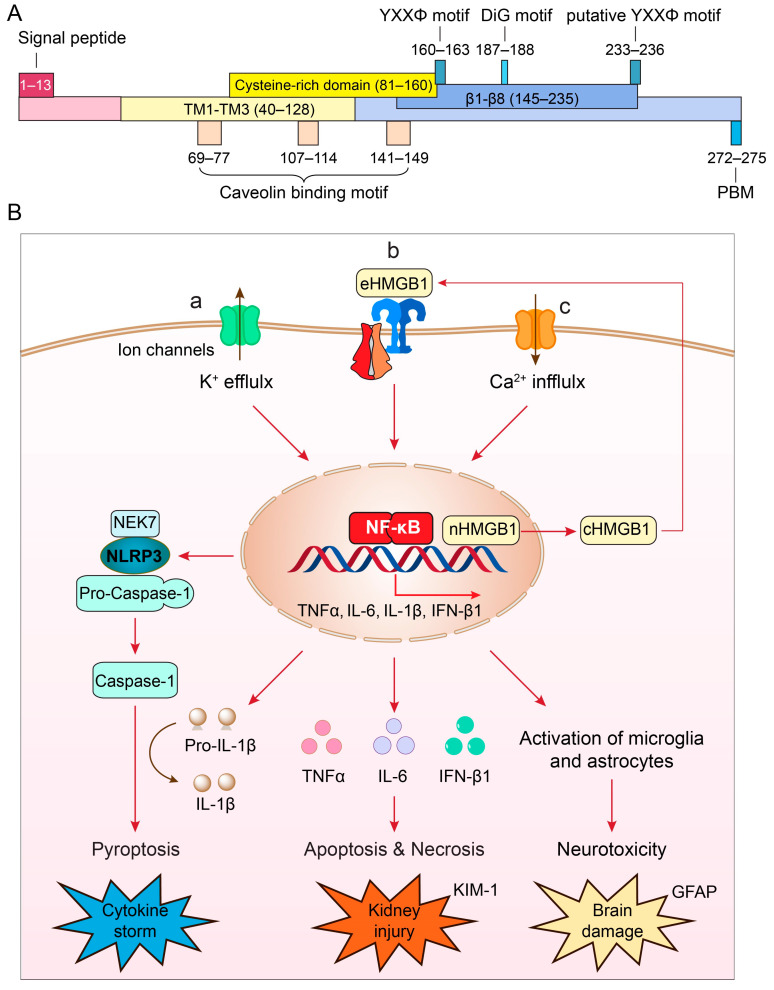
(**A**) Structure and unique features of SARS-CoV-2 ORF3a protein (adapted from [18,28]). The nucleotide sequence of *ORF3a* is 825 bp in length and encodes a 31 kD protein of 275 amino acids. It shows an extracellular N-terminal signal peptide (aa1–13), 3 transmembrane (TM) domains (aa40–128) that are across the cellular membrane and 8 antiparallel β-sheets (β1–β8, aa145–235) at the cytoplasmic C-terminus. Other functional motifs include 3 caveolin-binding motifs (aa69–77, aa107–114 and aa141–149) [32]; a YXXΦ motif (aa160–163) and a diG motif (aa177–178) [31,33,34], and a PBM (aa272–275) [18,35]. A putative YXXΦ motif (aa233–236) is also indicated [36]. (**B**) Cellular pathways of ORF3a-mediated effects that lead to the NLRP3 inflammasome activation and cytokine storm (**a**) [37,38,39], NF-κB-mediated cytokine production, apoptosis and necrosis leading to kidney injury (**b**) [40], and disruption of autophagy-lysosomal pathway by ORF3a that causes reactive microglia and astrocyte activation leading to neuroinflammation and brain damage (**c**) [41]. Elevation of kidney injury molecule 1 (KIM-1) is a specific biomarker for kidney injury [42]; elevation of glial fibrillary acidic protein (GFAP) is a well-accepted biomarker for brain damage [43,44].

The subgenomic *ORF3a* sequence is transcribed within the viral replication-transcription complex (RTC) of the double-membrane vesicles (DMVs) [18,45]. DMVs, formed by the rearrangement of the endoplasmic reticulum (ER), are also impacted by ORF3a, as it disrupts ER organization when ORF3a is overproduced [46]. Consequently, ORF3a plays an important role in the formation of ER-derived compartments such as DMVs that are associated with viral replication and transcription in host cells [46]. After protein synthesis of SARS-CoV-1 and SARS-CoV-2 ORF3a in the ER (32), ORF3a is transported from the ER to the Golgi apparatus, where it undergoes posttranslational modification via O-glycosylation before insertion into the plasma membrane and endomembranes, including endosomes and lysosomes [31,47,48,49]. This delineates a dynamic intracellular movement process for the ORF3a protein following its production. Functional domains within ORF3a, such as the YXXΦ motif (aa160–163) and the diG motif (aa187–188) in the cytoplasmic region, drive the transport of ORF3a from the ER to plasma membranes and other endomembranes [31,33,34,47].

ORF3a serves as a viroporin [18,28,29,50,51,52]–a viral transmembrane protein demonstrating ion channel properties within cell membranes [53]. Viroporins typically exhibit ion channel activities that can impact cell membrane permeability, Ca^2+^ homeostasis, and membrane remodeling. Their primary functions often involve virion morphogenesis, viral entry, replication, and the release (egress) of viruses [53]. Specifically, SARS-CoV-2 ORF3a forms a nonselective calcium (Ca^2+^)-permeable cation channel within a liposome system [28]. This channel allows the passage of large cations including NMDG+ and YO-PRO-1, resembling other Ca^2+^-permeable channels such as Trpv1, Trpa1, and P2x7 [28]. Furthermore, ORF3a may act as a non-selective cationic channel featuring a large pore and high single-channel conductance. Notably, a nonselective cation channel inhibitor, ruthenium red, blocks ORF3a-mediated ion conductance with an IC_50_ of 90±10 µM, demonstrating a distinctive mode of action compared to other known channels [28]. Notably, there are arguments challenging the notion that ORF3a functions as an ion channel protein by itself [54]. However, this does not dismiss the possibility that ORF3a might interact with other ion channel proteins to form a novel ion channel. These discrepancies suggest that while ORF3a might not be an ion channel itself, it could still influence function of the host cell ion channels, implying a role in ion channel modulation despite not being directly involved in channel formation.

The ORF3a protein exhibits several well-conserved functional motifs (Figure 1A), allowing us to hypothesize about its potential functions. Notably, it contains a TNF receptor-associated factor 3 (TRAF3)-binding motif spanning aa36–40 in both SARS-CoV-1 and SARS-CoV-2 [37,38,47,55]. Interestingly, TRAF3 plays a crucial role in regulating NF-κB signaling and cell death in SARS-CoV-1. Initial reports suggested that the interaction between ORF3a and TRAF3 through this binding site was necessary for the activation of NF-κB and the secretion of the cytokine IL-1β [37,38]. However, studies on SARS-CoV-2 indicated that deletion of this TRAF3-binding motif in the ORF3a protein did not affect its ability to induce NF-κB activation or NF-κB-mediated cytokine production [47,55]. The underlying cause of this discrepancy is presently unclear.

ORF3a harbors three putative caveolin-binding motifs located at aa69–77, aa107–114, and aa141–149 [30,32]. These motifs potentially regulate the intracellular trafficking of ORF3a [30,32]. Caveolins, membrane proteins involved in caveolae formation (small invaginations of the plasma membrane), might influence endocytosis during viral entry. Alternatively, these interactions could affect the sorting and transportation of the ORF3a protein to or from caveolar-carrying compartments within the cell. However, no specific reports have yet been published to delineate these potential functions.

Early mutational analyses, including our own, have pinpointed specific functional domains within the ORF3a protein, notably the YXXΦ motif (aa160–163) and the diG motif (aa187–188) situated in the cytoplasmic region. These domains play pivotal roles in directing the intracellular transport of ORF3a from the ER to lysosomes [31,33,34]. The YXXΦ motif, where Y signifies tyrosine, X represents any amino acid, and Φ denotes a bulky hydrophobic residue such as leucine or phenylalanine, is a recognized tyrosine-based sorting motif crucial for mediating the intracellular trafficking of proteins within the cell. On the other hand, the diG motif that we discovered comprises two conserved glycine residues at aa187–188 among mammalian CoVs [28]. Our study further indicated that disrupting either of these glycine residues, by deleting one (dG188), deleting both (dG187/dG188), or by substitution (G188Y), resulted in the retention of ORF3a in the ER and Golgi apparatus [31,47]. Structural analysis further suggested that the diG motif establishes connections to the YXXΦ motif through hydrogen bonds between two monomers, forming a hand-in-hand configuration that could facilitate ORF3a dimerization [31]. Thus, the YXXΦ motif and the diG motif in the cytoplasmic region likely drive ORF3a’s intracellular transport from the ER to plasma membranes and other endomembranes [31,33,34]. Interestingly, an in silico analysis of the ORF3a structure reveals another possible YXXΦ motif (YNKI) at aa233–236 [36].

The ORF3a protein possesses a highly conserved PDZ-binding motif (PBM) at its C-terminal end (SVPL; aa272–275) [18,35]. Given that there are more than 400 host cellular PDZ-containing proteins, the presence of the PBM in ORF3a suggests its potential to interact with a broad spectrum of host cellular functions [56,57]. It is a common tactic of pathogenic viruses, including SARS-CoV-2, to target cellular PDZ proteins [58]. Notably, the SARS-CoV-2 envelope (E) protein also carries a PDZ-binding motif. Both ORF3a and E proteins not only bind to some of the same cellular proteins [56] but also share similar functionalities. They are transmembrane proteins that can act as viroporins and participate in lysosomal deacidification, ultimately aiding in the release of progeny virions [18,59]. The PBM of SARS-CoV-2 E protein is involved in viral pathogenesis as manifested by its contribution to viral replication and lung edema [60]. Interestingly, the presence of ORF3a becomes crucial for viral reproduction in the absence of the E protein. SARS-CoV and SARS-CoV-2, lacking both the E and ORF3a proteins, are non-viable in Vero E6 cells or infected BALB/c mice [57,61]. The specific role of the PBM of ORF3a has not yet been fully characterized but it could potentially be a target site to block the interaction of ORF3a with host cellular proteins.

## 4. Role of ORF3a in Viral Production

ORF3a plays a pivotal role in multiple processes contributing to viral pathogenesis, particularly in viral assembly and release, which are crucial steps in the viral life cycle. SARS-CoV-2 ORF3a, but not SARS-CoV-1, facilitates viral release via the lysosomal exocytosis pathway [62]. ORF3a promotes this process by facilitating the trafficking of lysosomes to the plasma membrane and the fusion of exocytosis-related SNARE vesicle fusion proteins. This is achieved by aiding in the lysosomal targeting of the BORC-ARL8b complex [62]. Notably, SARS-CoV-2 ORF3a-mediated lysosomal exocytosis necessitates the activity of the Ca^2+^ channel TRPML3 [62]. Cells producing SARS-CoV-2 ORF3a demonstrated elevated cytosolic Ca^2+^ concentration compared to control cells, and TRPML3 knockdown impedes ORF3a-mediated lysosomal exocytosis [62]. Additionally, viruses such as mouse hepatitis virus (MHV), which lack ORF3a or SARS-CoV-1, exhibit reduced efficiency in releasing viral particles [62,63,64]. A recent report indicates that overproducing ORF3a facilitates viral release by counteracting BST2/tetherin, an antiviral host restriction factor. ORF3a achieves this by disrupting retrograde traffic, leading to the displacement of BST2/tetherin away from biosynthetic organelles. Tetherin, an interferon-induced protein, forms homodimers that link viruses to the surface of infected cells, thereby inhibiting virus release [65]. Note that another accessory protein, ORF7a, also antagonizes BST2/tetherin through direct protein-protein interaction [66].

ORF3a serves as a potent virulence factor contributing significantly to viral pathogenesis. Deletion of ORF3a from the SARS-CoV-2 genome results in a considerable reduction in viral production within human cells or transgenic mice [67,68,69]. Although ORF3a is not the sole contributor to viral production, combined deletion of ORF3a with other accessory proteins, ORF6, 7, and 8 (Δ3678), significantly weakens the virus, making it a potential candidate for a live attenuated virus vaccine [70]. Notably, while the Δ678 virus (without ORF3a deletion) had a mild effect, the Δ3678 virus exhibited a 3-log reduction in virus yield compared to the wildtype virus, underscoring the substantial contribution of ORF3a to the decrease in viral production [69,70]. Consistent evidence from studies, such as a CoronaVac vaccine investigation using inactivated virus, demonstrates a significant increase in antibody titers for ORF3a at two weeks post-COVID-19 diagnosis [71]. Additionally, a single deletion of ORF3a from the viral genome results in more than a 3-log reduction in virus production [72], corroborating findings from other investigations [67,69,73]. These collective studies underscore the critical role of ORF3a in viral production, emphasizing its potential as a promising antiviral target.

## 5. Role of ORF3a in Viral Pathogenesis: Induction of Cytokine Storm and COVID-19 Severity

The onset of a cytokine storm is a primary contributor to COVID-19-related morbidity and mortality [37,51,74,75,76]. These storms involve an excessive release of pro-inflammatory cytokines and chemokines, leading to hyperinflammation, acute respiratory distress syndrome, and multi-organ failure, often resulting in fatality. Overactive or dysregulated NLRP3 inflammasome activation in SARS-CoV-2-infected patients is linked to triggering cytokine storms, causing tissue damage and severe organ failure in individuals with critical COVID-19 [77,78].

The ORF3a protein has been implicated in the observed cytokine storms in severe CoV infections (Figure 1B-a) [39,79]. Specifically, ORF3a induces a pro-inflammatory immune response in infected cells, activating the NLRP3 (NACHT, LRR, and PYD domains-containing protein 3) inflammasome [37,38,39,51]. This multimeric protein complex triggers the secretion of pro-inflammatory cytokines [77,80,81], resulting in a hyper-inflammatory response. Such data indicate the substantial contribution of ORF3a to the severity and fatality of COVID-19 [37,51,74,75,76]. Notably, in animal models, infection of K18 human angiotensin-converting enzyme 2 (hACE2) transgenic mice suggests that ORF3a significantly contributes to lung pathology, and its deletion reduces the risk of a cytokine storm [69]. Mechanistically, ORF3a-mediated activation of the NLRP3 inflammasome leads to caspase-1 activation, converting pro-IL-1β to IL-1β through proteolytic processing, ultimately triggering pyroptosis [51,82,83,84]. Pyroptosis, an inflammatory form of lytic cell death, aids in rapid virus clearance and enhances the host’s antiviral response [80]. Note that ORF3a protein is not the only viral protein contributing to the activation of the NLRP3 inflammasome. Both SARS-CoV-1 and SARS-CoV-2 E protein also activate the NLRP3 inflammasome [85,86,87].

The exact molecular mechanism underpinning NLRP3 inflammasome activation by ORF3a remains elusive, but several models have been proposed: (1) ion channel activity, in which ORF3a exhibits ion channel activity disrupting cellular ion homeostasis, particularly calcium (Ca^2+^) and potassium (K^+^), leading to intracellular ion imbalances that trigger cellular stress responses, mitochondrial dysfunction, and eventual NLRP3 inflammasome activation [39,51]; (2) mitochondrial dysfunction, in which ORF3a’s interaction with mitochondria might induce damage or dysfunction, resulting in the release of mitochondrial reactive oxygen species (ROS), which could activate the NLRP3 inflammasome via the NIM811-sensitive mitochondrial-permeability-pore (mtPTP) [39]; and (3) direct NLRP3 inflammasome activation, in which ORF3a might directly activate the NLRP3 inflammasome [18,39], acting via HMGB1 (high-mobility group box 1), a downstream effector of the NLRP3 inflammasome to induce caspase-1-mediated pyroptosis (Figure 1B-b) [78,88,89]. HMGB1 is a ubiquitous protein released by microglia or macrophages upon NLRP3 inflammasome activation, and it promotes TLR4- and RAGE (receptor for advanced glycation end products) receptor-mediated pro-inflammatory cytokine production [90,91]. Elevated serum HMGB1 levels in COVID-19 patients correlate with poor prognosis [92,93]. HMGB1 inhibitors, like glycyrrhizin, reduce ORF3a-induced HMGB1 release and production of pro-inflammatory cytokines, potentially inhibiting SARS-CoV-2 replication [89]. Although a direct role of ORF3a on the HMGB1 release is yet to be confirmed through SARS-CoV-2 infection studies, these findings nevertheless indicate that SARS-CoV-2 ORF3a-mediated HMGB1 release is associated with host cellular pro-inflammatory responses and viral replication [89], and that HMGB1 could potentially serve as a possible cellular target to ameliorate the ORF3a effect.

## 6. Roles of ORF3a in Kidney Injury, Neuroinflammation, and Long-COVID Complications

Among millions of COVID-related deaths in the U.S. and worldwide, between a quarter and half of patients died with kidney complications such as acute kidney injury (AKI) [94,95,96]. AKI is a sudden decline of the normal kidney function that ranges from minor kidney malfunction to complete failure that could result in death. Patients with acute respiratory failure often also have AKI and have the worst overall prognosis [97,98]. A similar percentage of kidney complications was also found among those critically ill patients with COVID-19 [99]. Nearly half of those with AKI did not fully recover at discharge [96]. Continued decline in renal function was also seen in post-COVID patients, suggesting a long-COVID effect [100].

Although AKI is a major contributor to COVID-19-related death [96], the underlying cause of death in SARS-CoV-2-related AKI remains elusive. Besides the lungs, SARS-CoV-2 also infects the kidney via the ACE2 and other receptors such as CD209L/L-SIGN, CD209/DC-SIGN [101] and BSG/CD147 [102] in renal proximal tubular epithelial cells (RPTEC), where it supports kidney integrity and function [103,104]. Both SARS-CoV-2 RNA and proteins have been detected in SARS-CoV-2-infected kidneys [99,105], and viral particles recovered from the kidney were able to infect nonhuman primate RPTEC [106]. Histopathological examinations, including ours, biopsy or postmortem kidney tissues, showed that acute injury in RPTEC is most common in COVID-19 patients with AKI [94,107,108,109]. Those with acute tubular injury are associated with oxidative stress and inflammation-mediated cytokine release such as TNFα and IL-6 [94,107,108,109], which results in apoptosis and necrosis [110,111,112]. Furthermore, ORF3a, either expressed alone or in the context of viral infection in a human proximal tubular HK2 cell line, triggers increased expression of kidney injury molecule 1 (KIM-1) [40], a well-established biomarker for kidney injury [42]. KIM-1 is minimally expressed in normal kidneys, whereas it is highly elevated upon RPTEC injury [42]. In addition, ORF3a also induces renal injury in zebrafish and mouse models [40]. Consistently, SARS-CoV-2 infection caused dose-dependent and severe kidney damage including tubular damage and focal tubular collapse in mouse models that mimics those observed in COVID-19 patients [112,113]. Furthermore, deletion of *ORF3a* from the viral genome in an animal study showed reduced tissue damage [69]. However, kidney tissues were not examined.

Mechanistically, ORF3a induces renal tubule injuries by activation of NF-κB and STAT3 (activator of transcription 3) signaling [40]. ORF3a promotes STAT3 activation through interaction with a ubiquitin E3 ligase TRIM59, by which it dissociates the phosphatase TC-PTP from binding to STAT3, presumably by protein degradation, and hence it inhibits the dephosphorylation of STAT3, leading to persistent STAT3 activation [40]. Another study showed that the interaction of TRIM59 with ORF3a is part of the cellular ER-associated degradation (ERAD) response, in which ORF3a-induced ER stress elicits a selective form of cellular autophagy, known as ER-phagy, that transpires ERAD to degrade ER-residing proteins including ORF3a [47]. These observations may explain how ORF3a contributes to renal cell and tissue damages and COVID-19-related AKI.

SARS-CoV-2 may also infects the central nerve system (CNS) [114], leading to various neurological complications associated with COVID-19, both short- and long-term [115,116,117]. A national cohort study on patients with COVID-19 showed that COVID-19-associated neurologic complications link to a higher risk of disability and death [118]. Specifically, the virus affects brain glial astrocytes and induces neuroinflammation and neurotoxicity [116,117,119]. Studies, including our own, have observed neuroinflammation and apoptosis in reactive astrocytes, indicated by increased glial fibrillary acidic protein (GFAP), an astrocyte-specific brain injury marker [43,44], in postmortem brain tissues from COVID-19 individuals (Figure 1B-c).

Mouse models further demonstrate that SARS-CoV-2 infection activates microglial cells and astrocytes in the brain, resulting in pro-inflammatory cytokine production and damage to glial cells [117]. Consistent with the findings in infected individuals, ORF3a expression causes rapid onset neurological impairment, neurodegeneration, and neuroinflammation, mimicking neuropathological features of COVID-19 [41]. Mechanistically, ORF3a expression impedes brain cell autophagy progression, leading to the accumulation of α-synuclein and glycosphingolipids associated with neurodegenerative disease [41]. These findings suggest that ORF3a expression in brain cells drives neuropathogenesis and is an important mediator of both short- and long-term neurological manifestations of COVID-19 [41].

Overall, these studies underscore the potential clinical significance of ORF3a as a therapeutic target. Targeting ORF3a could potentially alleviate ORF3a-induced kidney injury and mitigate neuropathogenesis, offering a promising avenue for therapeutic intervention in COVID-19 complications.

## 7. Discovery and Development of Inhibitor Drugs Targeting ORF3a Protein

The multifaceted roles of ORF3a in viral pathogenesis, cytokine storm induction, and tissue damage associated with severe COVID-19 and long-term complications make it an attractive target for potential antiviral drugs or treatments [18,120,121]. Exploring inhibitor drugs against ORF3a protein involves several strategic approaches (Figure 1B).

### 7.1. Testing Well-Established Druggable Cellular Targets of Key ORF3a Regulators with FDA-Approved Drugs and Small Molecule Inhibitors (SMIs)

ORF3a-mediated signaling pathways involve key cellular regulators such as NLRP3, NF-κB, and related cytokines (TNFα and IL-6), all well-known druggable cellular targets relevant to COVID-19 [17,18]. A promising approach is to repurpose FDA-approved drugs that target these regulators against ORF3a-mediated pathways. For instance, ORF3a activates NLRP3, triggering cytokine storms linked to tissue damage and organ failure in severe COVID-19 cases (Figure 1B-a) [51,82,83,84]. Several compounds have shown potential as anti-NLRP3 inhibitors in preclinical studies or early clinical trials. MCC950 (CRID3) emerged as a notable NLRP3 inhibitor, demonstrating efficacy in various inflammatory disease models [122]. Notably, MCC950 also inhibits ORF3a-mediated NLRP3 inflammasome activation [51]. Additionally, FDA-approved drugs like doxycycline, metformin, and ibrutinib also exhibit inhibitory effects on ORF3a-mediated inflammasome activation [51]. Cinnamaldehyde, a natural compound acting as an NLRP3 inflammasome inhibitor, has demonstrated efficacy in vitro and in vivo by reducing inflammatory cytokine and ROS production in activated T-cells [123]. Other drugs targeting the NLRP3 pathway have been investigated in clinical trials or considered for repurposing to manage cytokine storms in COVID-19. However, no specific drugs have yet received official approval or widespread acceptance as standard treatments for COVID-19-related cytokine storms. Nevertheless, these NLRP3 inhibitors hold promise for testing against ORF3a-mediated inflammasome activation (Figure 1B-a).

The FDA-approved drug glibenclamide (glyburide) targets the Sur1-Trpm4 ion channel to inhibit NLRP3 inflammasome activation in microglia, effectively mitigating neuroinflammation [124]. Given that ORF3a, acting as a viroporin, triggers NLRP3 inflammasome activation and microglial responses in the brain [41], exploring whether ORF3a initiates NLRP3 inflammasome activation via the Sur1-Trpm4 channel is of great interest. Investigating whether glibenclamide can specifically impede ORF3a-induced NLRP3 inflammasome activation presents a compelling avenue for potential interventions against neuroinflammatory responses linked to SARS-CoV-2 infections. Such studies could offer valuable insights into managing the neurological complications associated with COVID-19.

NF-κB stands as another significant target to counteract the effects of ORF3a (Figure 1B-b), being a pivotal regulator in pro-inflammatory signaling pathways, where NLRP3 inflammasome activation is facilitated through NF-κB [51]. Auranofin, an FDA-approved NF-κB inhibitor, has displayed promising outcomes in impeding SARS-CoV-2 infection and associated inflammatory damages [125], warranting further examination against ORF3a-mediated responses.

Considering that ORF3a stimulates NF-κB to induce cytokine production such as TNFα and IL-6 [17,47,126], and that elevation of TNFα and IL-6 are two strong and independent survival predictors of patients with COVID-19 [127,128], targeting these cytokines is also appealing. TNFα and IL-6 serve as well-established druggable cellular targets with several FDA-approved anti-TNFα or IL-6 drugs either suggested for COVID-19 treatment or in various clinical trials [129,130]. For instance, three FDA-approved TNFα inhibitor drugs (infliximab, adalimumab, and etanercept) have displayed encouraging clinical outcomes in managing COVID-19 [129]. Similarly, two FDA-approved IL-6 inhibitor drugs (siltuximab and clazakizumab) have demonstrated promising results in treating COVID-19 patients [130], which have been described in the NIH COVID-19 Treatment Guidelines.

HMGB1 emerges as another promising druggable cellular target to counteract the effects of ORF3a, given its role in promoting HMGB1 secretion by ORF3a, which subsequently activates NF-κB and NLRP3 inflammasome (Figure 1B-c) [18,39]. Indeed, inhibition of HMGB1 has shown significant promise in mitigating NLRP3 inflammasome activation, renal inflammation, and tissue injury in a mouse pyelonephritis model [131]. Glycyrrhizin, a natural compound functioning as an HMGB1 inhibitor, has demonstrated its ability to reduce ORF3a-induced HMGB1 release and subsequent activation of the NLRP3 inflammasome [89]. Additionally, in a study utilizing a mouse traumatic brain injury (TBI) model, glycyrrhizin exhibited potential in alleviating HMGB1-mediated cognitive impairment induced by the NLRP3 inflammasome in the late stages of TBI [132].

### 7.2. Structure-Based Design of ORF3a Inhibitors

The cryo-EM structure of ORF3a is established [28] yet identifying specific target sites for precise inhibition remains elusive. Initial molecular docking studies using in silico models have highlighted potential interactions between ORF3a and various macroheterocyclic compounds, including chlorin e6 and cationic porphyrin such as TmPyP4 [133]. Subsequent analyses using fluorescence and UV-vis spectroscopy confirmed the interaction of chlorin e6 and porphyrin with ORF3a protein [133]. An intriguing aspect is that TmPyP4 is known for intercalating into and stabilizing G-quadruplexes (G4), which are present in DNA and RNA sequences containing guanine blocks [134]. Remarkably, G4s play a regulatory role in the life cycle of numerous viruses, including SARS-CoV-2. TMPyP4 demonstrates robust antiviral activity by targeting SARS-CoV-2 G4s, effectively suppressing SARS-CoV-2 infection in models such as the Syrian hamster and transgenic mouse, leading to reduced viral loads and lung lesions [135]. Interestingly, putative G4s are also detected in the ORF3a RNA sequence. Preliminary tests revealed that TMPyP4 mitigates ORF3a-induced cell death as measured by an MTT assay [136]. However, given that G4s are also present in plasmid DNA sequences, the specificity of TMPyP4’s inhibition of ORF3a and its binding relationship with the ORF3a protein, as predicted by docking analysis, remains unclear. Despite these uncertainties, leveraging a structure-based design for an ORF3a inhibitor holds promise and could potentially serve as a viable approach for further exploration.

### 7.3. Cell-Based High-Throughput Drug Screening

Cell-based high-throughput drug screening stands as a promising strategy for discovering and developing anti-ORF3a inhibitors, offering several advantages: (1) it enables comprehensive searches across diverse drug libraries, enhancing the potential to uncover a broader spectrum of inhibitors; (2) the screening process is target-specific, focusing exclusively on ORF3a production; (3) unlike structure-based drug design, this method is functionally driven, allowing for the identification of various inhibitors, including novel types such as allosteric inhibitors that act on ORF3a regardless of their binding to the ORF3a protein; and (4) the process automatically filters out cytotoxic compounds from HTS drug screenings.

Apart from employing mammalian cell based HTS, yeasts like fission yeast (*Schizosaccharomyces pombe*) have also been utilized in antiviral drug screenings via HTS [10,137,138]. Fission yeast presents distinct advantages compared to mammalian cell systems. For instance, it is easily maintainable and exhibits rapid growth, making it particularly well-suited for extensive drug screening. Moreover, its utilization proves to be more cost-effective than employing mammalian cells. Furthermore, employing yeast HTS shares all the advantages of a mammalian cell-based assay.

The use of yeast in HTS does not necessitate mammalian complexities but requires a functionally conserved output to indicate the presence of ORF3a, which need not necessarily correlate with viral or host cell biology. As an illustration, HIV-1 protease (PR)-induced cell death in fission yeast has served as a metric for HTS, distinct from its PR enzymatic activity, used to screen for HIV-1 PR inhibitors (PIs) [138,139,140]. Importantly, the expression of PR not only induces cell death in both mammalian cells and fission yeast [141], but HIV-1 PR, as an enzyme, also specifically cleaves its natural HIV-1 p6/MA substrates in fission yeast [139,140]. Given that all FDA-approved HIV PI drugs are designed to directly target PR, they effectively inhibit HIV-1 PR-induced cell death in fission yeast, thereby facilitating the identification of these PIs in this organism [142,143].

The key of using fission yeast cell-based system to study ORF3a is that the measured ORF3a effect on cellular processes must be highly conserved between yeast and human cells, particularly in fundamental cellular events such as cell proliferation and cell death [10,144,145]. Indeed, evidence indicates a strong correlation between ORF3a-induced cell death across various organisms including fission yeast, mouse, and human cells [17,69,146,147,148]. Hence, similar to HIV-1 PR, ORF3a-induced cell death stands as a viable primary endpoint for anti-ORF3a drug screening via HTS. Furthermore, this functional conservation facilitates incorporating this endpoint in both fission yeast and mammalian cell-based model systems. This integration serves the purpose of subsequent verification in mammalian systems upon the identification of potential inhibitors.

As ORF3a represents a viral protein, any potential ORF3a inhibitors identified through HTS require further validation by testing within the context of viral infection. Various standardized protocols, including our own, have been developed and utilized for screening small molecule inhibitors against SARS-CoV-2 infection [13,149]. In addition, to discern the specific anti-ORF3a effect from other viral properties during viral infection, or to ascertain that an inhibitor identified through HTS targets ORF3a specifically, it is essential to conduct tests evaluating the inhibitory effect of an inhibitor against both a wildtype virus and a virus with an ORF3a deletion [69,72]. Simultaneously assessing these scenarios helps evaluate the contribution of ORF3a to the observed results in drug testing.

## 8. Challenges and Future Prospectives

A major challenge in developing therapeutic drugs targeting ORF3a is that, like the spike (S) protein, ORF3a continuously mutates, leading to new variants that could alter their impact on viral pathogenesis, disease severity, and post-COVID conditions [150,151]. Distinct ORF3a mutants have emerged in variant-of-interest (VOI) and variant-of-concern (VOC) categories as defined by the World Health Organization, some linked to increased COVID-19 mortality or severity [18,150]. For instance, the epsilon variant carried the Q57H mutation, also present in beta and iota variants as a double mutant with S171L and P42L [152]. Additionally, gamma variants displayed D155Y and S253P mutations. These mutations, particularly Q57H and S253P, were associated with severe outcomes [151]. A unique T223I mutation appeared in BA.2 of the omicron variants and remains in the BQ.1, BQ1.1, and XBB1.5 variants to date [153]. This genetic variability can potentially affect the efficacy of antiviral drugs against ORF3a, leading to the emergence of drug-resistant strains. Note that, besides *ORF3a* mutations, there are also other viral gene mutations in those VOIs and VOCs. While there is a noticeable correlation between certain ORF3a mutants and clinical outcomes [30,150,151,154,155] without specific and confirmatory tests within the context of viral infection, we cannot dismiss the potential contribution of other mutations in viral genes.

Despite these challenges, the clinical correlation between ORF3a mutations and COVID-19 severity emphasizes the importance of targeting this protein therapeutically [30,150,151,154,155]. Throughout the pandemic, approximately 10 distinct natural ORF3a mutant variants were identified in association with VOIs or VOCs, including Q57H (epsilon), Q57H/S171L (beta), Q57H/P42L (iota), S253P (gamma), S253P/D155Y (gamma), S26L (delta), and T223I (omicron) [18,152]. Research indicates that certain ORF3a mutants such as Q57H, S26L, D155Y, and T223I exhibit varying levels of cytopathic effects but share similar mechanisms inducing cellular stress, inflammation, and apoptotic cell death [17,18,31,150,156]. Therefore, an effective ORF3a inhibitor should demonstrate broad-spectrum inhibition against these clinically relevant mutants [150,151,154]. Identifying such an inhibitor requires testing its efficacy not only against wildtype ORF3a but also against a panel of VOI/VOC-related natural mutants [18,31,47]. Successful inhibition of most or all natural ORF3a mutants by a specific inhibitor would indicate its broad-spectrum capability. Conversely, failure to inhibit a functionally relevant mutant could indicate the mode of action of the inhibitor. Identifying ORF3a’s binding sites may pave the way for future structure-based drug design.

Given ORF3a’s role in both acute COVID-19 and long-term complications, an ORF3a inhibitor could potentially treat both phases. Long-COVID complications often result from cellular or tissue damage caused by ORF3a, such as AKI and neurologic complications. Downregulating key cellular regulators with repurposed FDA-approved drugs against NF-κB, TNF, and IL-6 holds promise [17,47,126]. For instance, ORF3a triggers NF-κB-mediated inflammatory cytokine production, which correlates with COVID-19 severity [127,128]. FDA-approved TNFα and IL-6 inhibitors have shown clinical promise in COVID-19 treatments [129,130].

During acute infection, anti-NLRP3 inflammasome drugs could mitigate cytokine storms, especially important in older individuals [51,123]. A direct ORF3a inhibitor capable of addressing acute and chronic effects would be ideal, although no such inhibitor is known currently. Hence, discovering and developing direct ORF3a inhibitors remains a crucial need.

Finally, although the pandemic seems to be subsiding, considering current trends and the virus’s mutability, coronaviral infections are likely here to stay or become endemic [157,158]. Given that ORF3a is a recently acquired viral protein unique to SARS-CoV-1 and SARS-CoV-2 and its crucial role in viral pathogenesis and COVID-19, developing new antiviral drugs against ORF3a remains essential for ongoing and future coronaviral infections [18,28].

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
