# Peer review of "SARS-CoV-2 ORF3a Protein as a Therapeutic Target against COVID-19 and Long-Term Post-Infection Effects"

_pathogens, 2024, doi:10.3390/pathogens13010075_

Round 1

Reviewer 1 Report

Comments and Suggestions for Authors

THe authors have comprehensively reviewed the literature on role of ORF3a in COVID-19 infection and severity along with currrent and prospect anti-viral molecules targeting ORF3a singling.

There only minor revision mentioned below:
1. Provide reference for line 57 starting with "While vaccines with....." and line 60 starting with "Promising antiviral drugs..."
2. Extra space in line 391 before "Other drugs..." and line 521 before "Therefore..."

Comments on the Quality of English Language

English is well written minor edits 

Extra space in line 391 before "Other drugs..." and line 521 before "Therefore..."

Reviewer 2 Report

Comments and Suggestions for Authors

Coronavirus (CoV) accessory proteins, although dispensable for virus replication, modulate virus-host interaction and, frequently, influence CoV pathogenesis. The accessory gene ORF3a, is one of them. This review is focused in the potential of SARS-CoV-2 protein 3 as a therapeutic target. The manuscript is well-written and organized, however, there are some issues that should be considered to increase the scientific soundness of the manuscript.

Specific comments:

1. One of the main concerns with this review is that, often, the proposed functions/interactions of protein 3a are based on overexpression studies, not confirmed in the context of viral infection. Similarly, in many cases, the protein 3a role is proposed but not demonstrated with scientific data, and this is not clearly stated in the manuscript. In addition, frequently, the authors mix information extracted from other CoVs with the information from SARS-CoV-2 (without clearly indicating the virus). This would be considered when making assessments on the roles of ORF3a. Some (but not all) examples:

(i) Lines 112-115. References 20 and 21 are on SARS-CoV infection, not COVID-19 patients

(ii) Reference 31 is on protein 3a overexpression studies, therefore there is not direct demonstration that a virus lacking ORF3a has impaired DMVs formation

(iii) Lines 145-151. Only reference 29 is related to SARS-CoV-2, the other four references are on SARS-CoV

(iv) References 17, 27 and 36 are all reviews.

(v) Lines 170-185. References 39, 40 and 42 are on SARS-CoV-1; only reference 41 is on SARS-CoV-2

(vi) Lines 241-242. References 56 and 58 are on MHV virus, reference 57 is on SARS-CoV-1 virus with no information on SARS-CoV-∆3a virus.

(vii) Lines 271-283. Most data (refs. 39, 40, 45, 70 and 78) were obtained with protein 3a overexpression studies outside the infection context. References 39, 40 and 78 are on SARS-CoV

(viii) Lines 328-347. There is no solid indication on a role for protein 3a on kidney damage. Please note that reference 46 is on overexpression studies outside the infection context. Similarly, no kidney damage was analyzed in mouse models (refs. 62 and 65)

(ix) Lines 245-247. Please note that the main tetherin-related SARS-CoV-2 protein involved in virus release is protein 7a, data confirmed in the context of virus (encoding or not 7a) infection (Martin-Sancho L. et al, 2021, Cell 81:2656-68). Reference 59 on the putative role of protein 3a in this process is based on overexpression studies outside the infection context

(x) Reference 80 is on overexpression studies, not clearly demonstrating a direct role of protein 3a on the SARS-CoV-2 effect on HMGB1 release

(xi) Line 529. As indicated in all comments from this reviewer, the role of ORF3a in acute and long-term COVId-19 is proposed/hypothesized but not demonstrated in most of the cases, due to the absence of ORF3a specific studies in the context of virus infection

2. Lines 120-121 and 544-545. The assumption that ORF3a is unique to SARS-CoV and SARS-CoV-2 is essentially incorrect. Please note that many CoVs, including human CoVs, include at least one accessory gene after S gene and before E gene. Therefore, ORF3-like encoded proteins exist in all human CoVs with structure and/or functions similar to SARS-CoV protein 3a. Some examples are MERS-CoV ORF5, HCoV-229E ORF4, HCoV-NL63 ORF3, or HCoV-OC43 ns12.9. In fact, these genes can functionally replace each other (i.e., Zhang R. et al, 2015, J Virol. 89:11383-95).

3. Sections 4 and 5. Please note that section 4 mostly deals with the role of protein 3a in virus production and not in pathogenesis. On the other hand, section 5 is the one related with pathogenesis. Of course, there could be a relation between virus production (virus titers) and pathogenesis, but not necessarily in the case of a gene that is a virulence factor. In fact, please note that when virus titers and cytokine production are both reduced (ref. 62), no clear role as virulence factor is demonstrated unless the molecular mechanisms are clearly studied in that context.

4. Sections 7.1 and 8. Please note that the data on protein 3a being directly related in some pro-inflammatory effects caused by SARS-CoV-2 infection, such as cytokine induction, comes from overexpression studies and therefore it cannot be ruled out that other viral components influence these pathways. Indeed, these has been extensively reported by many authors in the context of viral infection. Similarly, VOCs and VOIs contain multiple mutations in several genes, therefore, association between an ORF3a mutation and a role in infection can only be demonstrated when it is studied in the context of a virus containing just that mutation. These issues would be considered.

Minor comments:

1. Title and throughout the whole manuscript text. Please note that ORF (Open reading frame) refers to a gene and not a protein. Then, the term “ORF3a” should be used when referring to the gene, while the term “protein 3a” should be used when referring to the protein.

2. Lines 80-98. Including here the information on the previously mentioned Paxlovid and Molnupiravir would be informative, to classify them in one of the mechanisms of action described in the paragraph. Similarly, they are not mentioned in the drugs that have obtained authorization from regulatory agencies, and this information would also be included.

3. Lines 286-290. Please note that additional information on the role of viral ion channel proteins in NLRP3 infalmmasome activation, and their relationship with virulence, come from SARS-CoV E protein studies (Nieto-Torres J.L. et al, 2014, PLoS Pathog 10:e1004077; Nieto-Torres J.L. et al, 2015, Virology 485:330-9).

4. Lines 348-366. Please note that the possible SARS-CoV-2 infection of CNS is still under debate. Detection of viral RNA and proteins does not necessarily imply production of infectious virus, although it can trigger neuroinflammatory processes. The data on this issue are somehow noisy as: (i) in transgenic mice virus goes to the brain in some situations, but this is not clearly observed in other models more closely resembling the situation in humans. And (ii) many data come from overexpression studies in CNS related cells, as the one in reference 47.

5. Line 223. Please note that SARS-CoV-2 E protein PDZ has been recently involved in pathogenesis (Honrubia J.M. et al, 2023, mBio 14:e0313622).

6. Reference 136. Please, eliminate. References to pre-print publications should be avoided, especially when the manuscript has not been published in three years.

7. Line 376. “Subsection” can be eliminated from the title.

Round 2

Reviewer 2 Report

Comments and Suggestions for Authors

The manuscript has been improved attending to the reviewers' suggestions.